# Future Proteins: Sustainable Diets for *Tenebrio molitor* Rearing Composed of Food By-Products

**DOI:** 10.3390/foods12224092

**Published:** 2023-11-11

**Authors:** Andrea Lienhard, René Rehorska, Barbara Pöllinger-Zierler, Chiara Mayer, Monika Grasser, Simon Berner

**Affiliations:** Institute of Applied Production Sciences, Sustainable Food Management, University of Applied Sciences FH JOANNEUM, Eggenberger Allee 11, 8020 Graz, Austriabarbara.poellinger-zierler@fh-joanneum.at (B.P.-Z.); simon.berner@fh-joanneum.at (S.B.)

**Keywords:** mealworms, sustainability, protein source, insects, by-products, insect rearing, alternative proteins

## Abstract

Since the human population is continuously growing, sufficient food with low environmental impact is required. Especially, the challenge of providing proteins will deepen and insects can contribute to a more sustainable and efficient source of protein for human consumption. *Tenebrio molitor* larvae are highly nutritious and rearing mealworms is more environmentally friendly compared to the production of traditional livestock meat. To use *T. molitor* as a more sustainable alternative to conventional proteins, it is essential to apply diets from a local and sustainable source. Therefore, the objective of this study was to find local by-products or leftovers which can be used in mass production of larvae as a main substrate. Feeding trials investigating twenty-nine different substrates were conducted to evaluate larval growth performance and adult reproduction by determining development times, survival rates, biomass, and fecundity. Several suitable by-products were identified that can be used in high quantities as single component diet for *T. molitor* rearing, revealing a high survival rate, short development time, high mean total biomass, and successful breeding. The most successful substrate—malt residual pellets—was found to be an alternative to the most used substrate, wheat bran. Furthermore, corn germ meal, sweet chestnuts, bread remains, soybeans, sweet potatoes, and wheat germs have been discovered to be suitable diets for *T. molitor*. Moreover, the findings of this study contribute towards using several substrates as supplements.

## 1. Introduction

In light of the growing human population and the increasing demand for food, alternative and sustainable protein sources with low environmental impact can help to face this demand [1]. The production of traditional livestock meat is associated with detrimental environmental effects such as global warming, air and water pollution, or land degradation [2]. The yellow mealworm, *Tenebrio molitor* Linnaeus, 1758, however, can be reared in an environmentally sustainable manner, has excellent nutritional characteristics (high protein content, all essential amino acids), gives rise to low emissions of greenhouse gases, requires much less land than traditional livestock, has a high reproductive capacity, and displays a very high efficiency in food conversion [1,3,4,5,6]. To date, commercial insect rearing is an important industrial sector across Europe and, economically, mealworms are among the most promising insects for the industrial utilization and commercial large-scale production of sustainable proteins [7].

*Tenebrio molitor* is a coleopteran species belonging to the family Tenebrionidae. Larvae of the so-called Darkling Beetles are commonly known as yellow mealworms and are considered as a pest, primarily feeding on farinaceous materials [5]. Mealworm larvae undergo several molts until they reach the pupal stage and finally emerge as adult beetles. The duration of the larval stage varies from 57 days in controlled conditions [8] to 628 days in natural conditions [9], with an average duration of 112 to 203 days [5]. The pupal stage lasts from six [10] to 20 days [5,11].

Mealworms are becoming more and more important in sustainable protein production, but to make *T. molitor* competitive with other, more-traditional protein sources, mass production must be optimized [5]. The optimization of different parameters for production such as environmental and dietary conditions to ensure optimal growth, costs (especially by using low-cost food sources), or the rearing techniques (automated rearing) is necessary to achieve this goal [5]. In recent years, detailed investigations on environmental and physical conditions for mealworm rearing were published, and the most important factors like the optimal temperature of growth (25–28 °C and ≥70% relative humidity) are well studied [5,11]. Indeed, diet plays an important role in insect rearing and has been identified as one of the major challenges for the industry, influencing the growth, the development time, the number of instars, the survival rate, adult reproduction, and the mortality of all stages [2,5,7,12]. Initial research on the nutritional requirements of mealworms date back to 1950, figuring out that an optimal diet consists of 80–85% carbohydrates, 5–10% yeast, and the addition of B-complex vitamins [13]. In the absence of carbohydrates or vitamins of the B-complex, the growth of *T. molitor* almost stops; the addition of vitamins A, C, D, E, or K to the diet, however, provide no beneficial effect [14]. Although *T. molitor* can only be fed with wheat bran (it contains all the necessary nutrients) [15], diet supplementation is beneficial because nutrients do not appear in optimal proportions in most single substrates [5,12,14]. A diet composed of wheat bran in addition to a water source (fresh vegetables or fruits such as carrots, apples, cabbage, or potatoes) and a protein source (beer yeast, soy protein, casein) is the most common diet composition in the mealworm industry [5,7]. Further work was conducted in subsequent decades and some studies investigated the effects of different substrates (or supplements) on the growth, development, and chemical characteristics of mealworm larvae [1,2,3,7,16,17,18,19,20]. All these studies highlight a certain plasticity of *T. molitor* in relation to the substrate, with variations in the development times and nutritional values of the larvae. Although *T. molitor* is widely used for food production, certain aspects of rearing conditions are largely unknown and, furthermore, the high plasticity of larvae in growth is still a big hurdle in rearing. Growth performance is additionally affected by several factors such as temperature, humidity, photoperiod, population density, the strain, and food quality. Therefore, growth rates and development times are often not directly comparable [7,16]. It is thus not surprising that the results of feeding experiments in general show a great range of growth parameters like developmental time values [21].

A hurdle for the mass production of insects is the fact that it is still (too) expensive, so means are being sought to reduce costs. One possibility to keep costs for insect rearing low is the use of side streams or by-products [22]. Moreover, to use edible insects as a more sustainable alternative to conventional proteins, it is essential to apply substrates from a local and sustainable source. This can be achieved by rearing the insects on diets composed of local industrial (e.g., from food industry) or agricultural by-products [2]. A few studies attempted to increase sustainability by using diets composed of organic by-products for *T. molitor* [1,2,3,22,23,24,25,26].

The growth performances of larvae are heavily influenced by diet, although studies show different outcomes. Five different foodstuff products (brewery spent grains, bread and cookie leftovers, mixes of brewer’s spent grain or bread with cookies) were evaluated and it was shown that mealworms can be reared on leftovers and by-products with proficient outcomes, additionally allowing to reduce waste materials [1]. A further study also confirmed that mealworms can be successfully grown on diets composed of organic by-products—originating from beer-brewing, bread/cookie baking, potato processing, and bioethanol production—by investigating the growth performance and feed conversion efficiency [2]. A self-selection experiment revealed that the most suitable substrates for rearing *T. molitor* larvae were mainly wheat bran and flour, oat bran and flakes, maize hulls, rice flour, lupine flour, and potato flakes [22]. However, combinations of organic wastes (mixed vegetable waste, garden waste with green biomass, cattle manure, horse manure), showed high mortality rates of all combinations in pure form and with the supplementation of 10% chicken feed [23]. Five different spent mushroom substrates were examined in another study; consequently, young larvae did not survive on any substrate except the spent shiitake [24]. Four diets composed of food by-products, namely beet molasses, potato steam peelings, spent grains and beer yeast, bread remains, and cookie remains, were tested and significant differences in the feed conversion efficiency, survival rate, and development time were reported [3].

The majority of the aforementioned studies employed mixes of several substrates as diets. Data on the growth of mealworms on single component diets could improve the knowledge and can further contribute to the development of nutritionally balanced dietary mixtures for the mass production of *T. molitor* [7]. This is extremely important—authors have further stated—as *T. molitor* has a wide range of food preferences (>50 different substrates), so finding a mixture of suitable components is more demanding than for other species. Because of the fact that only the effects on combinations (for instance of different amounts of proteins, lipids, or carbohydrates) were investigated, it is very difficult to assess if mealworms could be reared on single substrates directly gathered from the producers in high quantities [1]. Mealworms have the ability to select food in order to balance their intake according to their nutritional needs [5,14]; if a mixture is applied, food which has not been eaten has no nutritional value. Nevertheless, reliable statements about the nutritional value are dependent on the knowledge of the amount of consumed food [27]. Moreover, by-products often occur in high amounts; thus, only disposal in high quantities is economically reasonable [28].

In this comparative study, twenty-nine different substrates were examined to detect a sustainable diet for the mass production and breeding of *T. molitor* to subsequently retain a sustainable protein source for food purposes. The aim of this long-term study was to find a single component diet which can be used to efficiently rear *T. molitor*. To ensure larval development and adult reproduction, feedings trials over several generations were conducted. Growth performances, survival and mortality rates, development times, reproduction rates, and the consumed feed of *Tenebrio molitor* larvae and adults on diets composed of various local by-products and leftovers were examined. Additionally, the nutritional and chemical composition of offered substrates was evaluated.

## 2. Materials and Methods

*Tenebrio molitor* larvae were obtained from a colony which had been reared since 2019 at the University of Applied Sciences (FH JOANNEUM GmbH, Graz, Austria). This colony was originally obtained from “abc-a better choice GmbH” on the 30 January 2019. All specimens were kept in a climate chamber at 27–29 °C with a relative humidity of 60–65% (KBF-S, WTC Binder GmbH, Tuttlingen, Germany). As recommended [21], specimens were kept in constant darkness. Freshly emerged larvae were raised in 100% wheat bran for 21 ± 2 days (depending on the eclosion time) and were then transferred to the different substrates (in earlier stages the mortality rate is too high to obtain valid results, cf. [2]). Before the start of the feeding experiments, all larvae were starved for 24 h. For the feeding trials, two boxes—8 × 8 × 5 cm for larvae and 15 × 8 × 5 cm for adult beetles (Curver^®^ Luxembourg)—of each diet were stocked with 50 individuals of *T. molitor* (two boxes per substrate in most cases). Pupae were transferred to ice cube trays and reared individually until adult emergence under the same environmental conditions as larvae and beetles. At the beginning of the experiments, 2–18 g (depending on the density) of each substrate was placed in each of the plastic boxes. All cultivation boxes containing the different substrates were stored in the same climate chamber under the same rearing conditions. Twenty-nine substrates were tested and thirty-one different feeding experiments were conducted (WBc = wheat bran with the addition of carrots; LSWBc = second strain of *T. molitor* obtained from “dieWurmfarm”, Carinthia, Austria). Carrots were provided three times per week. Larvae were allowed to feed ad libitum and fresh feed was provided weekly to avoid accumulation of feces and contamination of frass. The frass was separated from the remaining food by using different sieves. Substrates were placed in the climate chamber for 24 h (conditions see above) prior to use. Boxes were visually evaluated five times per week to check larval health, to determine mortality, and to remove dead individuals. Once per week all specimens were separated from the feeding substrate, counted, and weighed individually. An additional water source is essential to increase the fecundity and longevity of *T. molitor* and thus, 0.1 to 0.2 mL of water was provided with a pipette twice a week to each box. Figure 1 illustrates the setup of the feeding experiments.

### 2.1. Diet Selection and Preparation

Twenty-nine substrates were selected as experimental diets (for further information, see Appendix A). The majority of these by-product streams were locally available in high quantities, resource- and cost-efficient in mass rearing, free from any harmful contaminants, logistically easy to handle, and storable with a long shelf-life [28], and included the following: wheat bran (WB), malt residual pellets (MRP), corn germ meal (CGo), coffee grounds (CG), coffee chaff (CC), pumpkin seed oil cake (PSO), pearl oyster mushroom mycelia with coffee grounds and coffee chaff (AK), *Fallopia* x *bohemica* (FB), Sida (Si), sweet chestnuts with peel (CN), brewer’s spent grain (draff, D), garlic peel (G), bread remains (B), runner beans (Be), Mur sand (sand), hempseed oil cake (WC), acron flour (EH), sawdust (SD), Soybeans (SO), *Urtica* (U), foam peanuts (V), pearl oyster mushrooms (A), wheat straw (WS), sweet potatoes (SP), potatoes (Maestro, Musica, Fries), jerusalem artichoke (Papas), and wheat germ extracts (WK). Two control boxes were equipped with wheat bran and carrots (control diets: WBc, LSWBc) and one with no food provided (control group). The following substrates were dried using a drying cabinet (TS540WTC Binder GmbH, Tuttlingen, Germany) at 50–70 °C for 24 h before use: U, CG, AK, FB, D, B, S, A, SP, Papas, potatoes (Maestro, Fries, Musica), and WK. MRP, CC, G, B, Be, WC, SO, V, WS, SP, potatoes (Maestro, Musica, Fries), and Papas were ground (Vitamix, VM0105E, VITA-PREP^®^ 3, Frankfurt, Germany). After drying, *Urtica* leaves were manually crushed. Brewer´s spent grain was additionally frozen at −20 °C prior to use.

### 2.2. Data Analysis

The following values were calculated/determined:Survival rate of larvae;The survival rate was determined by daily counting the number of dead larvae for each treatment and calculating the rate in percentage based on the total number of larvae.

2.Development times (larvae -> pupae, pupae -> adults);Development time of larvae was recorded beginning with the first day of the experiment (with about three-week-old larvae) until the first pupa emerged.

3.Individual weights of the larvae, pupae, and adult beetles;Pupae and beetles were weighed shortly after eclosion. For boxes containing > 10 larvae, ten randomly chosen larvae per box were weighed individually once a week and placed back to the box. For boxes holding < 10 larvae, all specimens were weighed individually.

4.Total biomass of larvae, beetles, and pupae;5.Percentage of emerged pupae (pupation rate) and emerged beetles;The emergence of pupae and beetles was calculated based on the total number of initially investigated larvae and is given as a rate.

6.Sex ratio [%];Sex was determined for pupae and was subsequently checked again for adult beetles to assure the correct determination.

7.Anomalies of beetles (deformed adults in %);Adult beetles were visually inspected for any abnormality or deformation (any defect in appearance or mobility).

8.Initial weight of feed and amount of remaining feed (= FC, “feed consumed”);FC = subtracting the remaining substrate from the total amount of the substrate provided (initial weight of feed) [7].

9.Chemical composition of the offered diets;Substrates and larvae were analyzed by “Futtermittellabor Rosenau” (Futtermittellabor der Landwirtschaftskammer Niederösterreich, Lower Austria), following the VDLUFA regulations (Association of German Agricultural Analytic and Research Institutes, https://www.methodenbuch.de/produkt/methodenbuch-band-iii-futtermittel/, accessed on 10 November 2023). The total content of protein was determined using the Kjeldahl method. Depending on the substrate, the fat content was analyzed by two different methods. For all substrates, the extraction was performed with petroleum ether as solvent. Substrates of vegetable origins containing soy protein, yeast, or potato protein were treated with hydrochloric acid before the extraction. To determine the crude ash content, the samples were burned at 550 °C and weighed afterwards. To verify the crude fiber content, the defatted substrates were treated with sulfuric acid and potassium hydroxide solution, filtrated, washed, dried, weighed, burned, and the residue was weighed again. For the sugar content, samples were dissolved in ethanol, cleared with Carrez solutions, and the content was determined by the Luff Schoorl method. CGo (corn germ meal) was analyzed by BOKU (University of Natural Resources and Life Sciences, Vienna). Larvae for chemical analysis (“TM”, fed with WB and carrots) were fasted 24 h before being killed by freezing at −18 °C and were also analyzed by “Futtermittellabor Rosenau”.

### 2.3. Statistical Analysis

A Kruskal–Wallis test was conducted because data on the individual weight of larvae [g] in comparison to the substrate were not normally distributed. Box 1 and box 2 of the substrates was treated independently. A post hoc analysis, namely Dunn’s test, was used to verify statistically significant differences among feeding trials. Dunn’s test is a multiple comparison test and provides *p*-values for pairwise comparison between groups. Differences among feeding trials were considered significant at *p* < 0.05. Statistical analysis was performed using R version 4.3.1 (https://www.r-project.org/, accessed on 10 November 2023).

## 3. Results

Single substrates were divided into the three following groups based on the success in feeding trials to ease data handling beforehand: eight “successful” single substrates (obvious weight gain and pupation of larvae), seven “improvable” (initial weight gain, no pupation), and fourteen “not recommended” substrates yield to no weight gain and show a high mortality rate.

### 3.1. Survival Rates of Larvae Grown on Different Substrates

As illustrated, two mono substrates (WB and MRP) show a survival rate of >97% for *T. molitor* larvae (Figure 2). Considering the control group (WBc), all larvae (100%) survived until the first pupa was observed. For sweet potatoes (SP) and chestnuts (CN), there was a survival rate of 69% and 76%, and for bread remains (B) a rate of 34% was observed. Mortality was highest for corn germ meal (CGo) and soybeans (SO); only a small percentage (≤25%) of larvae survived until the first pupa emerged.

Survival rates of larvae reared on “improvable” substrates are shown in Figure 3. Regarding those substrates, larval survival rate continuously decreased with time, even though some individuals were able to survive long periods. Two individuals fed with *Urtica* and pearl oyster mushrooms, respectively, survived for more than 440 days and one larva fed with EH for 551 days. A vast decrease in the number of individuals was recorded for AK, with a survival rate of <10% after 28 days, although a single larvae survived for 415 days. The fastest decrease in survival, with a mortality of 100% after 145 days, was revealed for larvae fed with draff.

The survival rate was low for saw dust (SD) and wheat straw (WS). Within 15 days, the majority (99%) of individuals died (Figure 4); therefore it is assumed that these substrates are harmful to larvae. Sida and *Fallopia* also showed a strong decrease in larval survival, with a survival rate of less than 10% after 40 days (1% after 56 and 77 days, respectively).

High mortality was furthermore observed for individuals reared on the following eight substrates: CC, PSO, Musica, Be, G, CG, V, and sand (Figure 5), with an overall survival period of less than 200 days. Within the first month, the survival rate of individuals fed with CC, sand, V, CG, and G was <10%. The “control” group (with no food provided) survived for 53 days, with a vast increase in mortality after day 13. Regarding CC, G, sand, V, and CG, the mortality was even higher as compared to the control group, indicating harmful contents in these substrates.

### 3.2. Effect of Substrates on the Individual Weight and the Total Biomass of Larvae

The individual larval weight for each successful dietary treatment is provided in Figure 6. Larvae fed with control diets—wheat bran and carrots (LSWBc and WBc)—showed high mean individual weights, reaching 0.09 g and 0.14 g, respectively. Except for MRP with a mean individual weight amounting to 0.13 g and bread with only 0.03 g, the mean individual weight of larvae ranged between 0.06 g and 0.08 g. The mean individual larval weight at the initiation of the feeding trial was 0.0033 g.

A significant difference between feeding trials based on the Kruskal–Wallis test was found. For the eight successful substrates, the boxplots obtained from the Kruskal–Wallis test are shown Figure 7. The comparison of different boxes of the same substrate (box 1 and 2) was not evaluated as statistically significant (*p*-values are shown in Appendix A, Appendix A).

The mean total biomass of larvae reared on ten different substrates assumed to be “suitable” is depicted in Figure 8. The most intensive increase in biomass was achieved by larvae fed with the control diets (LSWBc and WBc) and the highest mean total biomass was recorded for larvae reared on MRP with 7.12 g, followed by the control diet (WBc) with 7.11 g. The second control diet and WB amounted to a maximum of 6.65 g of total mean larval biomass. As only larvae were weighed, for four feeding trials an abrupt decrease in total larval biomass was noted, starting with the emergence of the first pupa. For all other substrates (CN, WK, SP, B, CGo, SO), the mean total biomass was found to be much lower. Although larval development time was shorter for the control diets, the total biomass was comparably high for two more single substrates, namely wheat bran and malt residual pellets.

The mean total biomass of larvae grown on “improvable substrates” is provided in Figure 9. An initial biomass growth was recorded for eight substrates, although biomass rapidly decreased with time for all substrates (AK, EH, D, WC, A, U, Maestro). Only Maestro displays a slight increase, and the highest mean total biomass was recorded for *Urtica*, reaching 1.06 g on day 221 of the experiment. The mean total biomass for “not recommended” substrates are shown in Appendix A (Appendix A). A high decrease in weight without any weight gain was recorded for those fourteen substrates. 

### 3.3. Growth Parameters and Development Times of Pupae and Beetles

The development time (emergence of the first pupa) was the shortest for one of the control diets (WBc) with 42 days, and for MRP and WB the development time reached 52 and 53 days, respectively (Table 1). Surprisingly, the first individual of the second investigated strain (also fed with wheat bran and carrots, LSWBc) showed a much longer development, with the first pupa appearing after 73 days. For all other substrates, the development time was prolonged with 106 to 227 days. Pupation rate of control diets reached 96–98%, and the highest rate for a single substrate was recorded for MRP (93%), followed by WB with 79%. All other single substrates showed low pupation rates of <35%. The pupation rate of CGo, SO, and B was very low, at only 2–3%. Only one substrate—except for the control diets LSWBc and WBc (98%)—namely MRP, had a high rate of emerged beetles, amounting to 95%. The mean pupal weight reached from 0.11g ± 0.00 to 0.17g ± 0.03, with a minimum of 0.06 g (WK) and a maximum of 0.21 g for MRP and WB (0.23–0.24 g for control diets). The sex ratio was around 50/50 (m/f) for all substrates, except WK (41/59) and CN (43/57). As only three pupae emerged from CGo and B (ratio 33/67 and 67/33, respectively), no considerable statement can be made.

The mean individual weight of beetles was highest in MRP and in one of the control diets (WBc), with 0.14 ± 0.02 g, and the highest individual weight of a single beetle was measured for the control diets and MRP (with 0.19–0.21 g), respectively (Table 2). The number of days from pupation until eclosion was not influenced by the substrate and was on average stable, with six to seven days for all feeding trials. Not a single morphological abnormality was found in beetles fed with wheat germs (WK), whereas the rate of abnormal beetles was very high, amounting to 83% deformed individuals fed with chestnuts (CN).

### 3.4. Feed Consumed

By subtracting the remaining substrate from the total amount of the substrate provided (initial weight of feed), the FC was calculated for suitable substrates. According to the results concerning this study, MRP showed the highest amount of consumed feed (Figure 10) and the sharpest increase in FC, followed by WB.

### 3.5. Nutrient Composition of Substrates

The starch content of successful substrates ranged between 17.8% for wheat bran and 60.4% for bread remains (Figure 11). The protein content varied from 8% (chestnuts) to 38.2% (wheat germs). The substrates which were not successful in feeding trials had higher amounts of other nutrients than carbohydrates like proteins or fiber. Although runner beans showed a high starch content (40.4%), they were considered as “not suitable” for *T. molitor* rearing. More detailed information on (micro-) nutrients is given in Appendix A Appendix A.

## 4. Discussion

As shown in previous studies, diet greatly affects the whole life cycle of *Tenebrio molitor*, concerning for instance the development time, the survival, or the fecundity, and is consequently considered as one of the most important factors in rearing [1,2,3]. In this study, vast differences among substrates in all tested growth parameters were found (cf. Table 3). Based on this performance, eight local and sustainable by-products suitable as a single or main substrate for *T. molitor* rearing were identified, namely (i) wheat bran (WB), (ii) malt residual pellets (MRP), (iii) corn germ meal (CGo), (iv) sweet chestnuts (CN), (v) bread remains (B), (vi) soybeans (SO), (vii) sweet potatoes (SP), and (viii) wheat germs (WK). Six single substrates furthermore led to successful breeding cycles. These diets (WB, MRP, CGo, CN, SP, WK) still serve as main, single rearing substrates for several generations. The F2 generation was investigated in a study to evaluate the safety of mealworms as a sustainable protein source in human nutrition [29]. An analysis of the microbiome of *T. molitor* showed that the substrate had no negative effect on the microbial load of *T. molitor*, allowing a risk-free human consumption of larvae fed with these by-products [29].

These eight aforementioned “successful” local by-products led to the growth of larvae and consequently to pupation and can be used as main substrates. However, growth rates, development times, fecundity, et cetera, may be improved by adding specific supplements. Many authors are in agreement that one of the most beneficial supplements in mass rearing is the addition of beer yeast at concentrations of 5–10% as it contains important growth factors for yellow mealworms and contributes to a shorter development time, a higher survival, and maximizes weight gain [2,3,5,13,14,16,22]. Another frequently used supplement is the addition of carrots. The addition of carrots as a water source resulted in an increase in the survival rate (≥80%) and reduced the development time from 145–151 days to 91–95 days [3]. The increased survival and shortened development time is also due to carrots functioning as an additional water supply [13]. Although *T. molitor* can survive under extremely dry conditions for extended periods of time and is able to obtain water from the atmosphere and from the food ingested, larvae grow faster in humid conditions (>70% RH) and when there is any source of water [3,5]. Once suffering from water deprivation, larvae ingest less food and development halts [5]. Larvae reared on several substrates in this study targeted the supplied water source and specimens were observed to actively drink water (e.g., CG, CC, PSO, B, CN, WC, U).

Wheat bran is the most used and studied diet for mealworm rearing as a main substrate with a water source and a protein source like beer or dried yeast, casein, albumin, zein, lactalbumin, etc. [5]. To our knowledge, malt residual pellets, corn germ meal, wheat germs, and sweet chestnuts have not been used as single substrates in *T. molitor* feeding trials. Nevertheless, those substrates appear to be a good alternative to wheat bran-based diets. Even though wheat products are produced in Europe throughout the year, the amount produced during winter is less [28]; thus, alternative substrates like MRP can be used. Malt residual pellets showed a high survival rate (99%), short development time (52 days), high mean total biomass (7.12 g), high mean individual larval weight (0.13 g), and high food intake. Sweet potatoes were considered as a suitable main substrate for *T. molitor* in our feeding experiments and were also recommended in a study as the only suitable by-product for rearing, out of five different plant based raw materials, concerning the results of growth and survival [25]. Sweet potatoes are suitable since they are rich in carbohydrates and β-carotene. Surprisingly, soybeans also led to pupation (rate of only 2% and no emerging beetles), even though it was shown that soybean meal was one of the least consumed components in a self-selection experiment [26]. Soybeans, although rich in proteins, contain a trypsin inhibitor that can negatively influence larval growth [30]; thus, growth is restricted in most legume flours [7]. Bread remains were obtained from local bakeries and were variable in their composition (bread rolls, dark bread, pastries, cookies, etc.). With a pupation rate of only 3% and no emerging beetles, this substrate obviously has to be optimized. According to the literature, larvae fed with bread, cookies, or a mixture of both did not reach the pupal stage after one year of rearing and the presence of cookies in the diet even worsened the growth performances and induced a decrease in humidity in the larval body [1]. Mealworms fed only bread did not convert the nitrogen of this substrate efficiently into body mass [3]. Additionally, spices such as cinnamon in bread remains can be toxic to insects [2].

Seven substrates led to an initial growth and may be suitable as a partial substitute added to other main substrates, namely brewer´s spent grain (D), hempseed cake (WC), acron flour (EH), pearl oyster mushroom mycelia with coffee grounds and coffee chaff (AK), pearl oyster mushrooms (A), potatoes (Maestro), and *Urtica* (U). In most studies, potatoes were added as a supplemental diet with positive effects on the growth rate due to the high carbohydrate and low fiber content [22,26]. Compared to the control group (100% WB), dry potatoes significantly improved development time, survival, growth, and fecundity [14]. Fresh potatoes are furthermore attractive vegetables to *T. molitor* adults because of the high content of moisture (77–85%) and were used as a trap or sampling tool for monitoring purposes [31]. Considering our experiments, however, potatoes as a main diet had negative impacts on the growth rate, larval survival, and the development. Three different varieties of local potatoes were tested and, surprisingly, showed congruent results. Only one variety (Maestro) led to minimal growth, in contrast to the other two varieties, namely Musica and Friesländer. This is why potato starch is more resistant to digestion by Tenebrionidae than starch from wheat or maize for instance [32]. Furthermore, potato glycoalkaloids can have a toxic effect on insects like mealworms that do not consume potatoes in nature [2]. Different contents of brewer’s spent grain were mixed with wheat bran in several studies [18,33]. In contrast to the results gained in our study, 100% brewer´s spent grain showed no differences in development times, larval weight, or survival rate compared to lower ratios or the control diet (100% WB) [18,33]. Moreover, pupation rate was above 90% for larvae grown on all ratios, and brewer´s spent grain as a single substrate gave the highest pupation rate of 100% [18]. Spent substrates from different mushrooms are highly variable in physical, chemical, and biological properties, and each mushroom species has its own specific utility [24]. Most studies revealed negative effects on growth parameters for larvae fed with mushrooms. By replacing wheat bran or rice bran with 20–60% spent shiitake, larvae survived, but the larvae of all treatments failed to pupate [24]. By substituting 40–50% of wheat bran with a mushroom substrate, larvae were lighter and required longer development periods [17]. The herein used pearl oyster mushroom can be used as a partial substitute for conventional feed as this substrate (in small amounts) presumably does not negatively affect the survival and development of larvae.

Fourteen “not recommended” substrates were identified, showing low survival rates and halting growth, namely jerusalem artichoke (Papas), potatoes (Musica, Friesländer), wheat straw (WS), sawdust (SD), runner beans (Be), garlic peel (G), Sida (Si), *Fallopia* x *bohemica* (FB)*,* pumpkin kernel cake (PSO), coffee chaff (CC), Mur sand (sand), foam peanuts (V), and coffee grounds (CG). These substrates may comprise harmful substances or contain ingredients that have an adverse effect on growth due to the imbalance in macronutrients [26]. In fact, the carbohydrate content was low in those substrates, except for beans (Figure 11). The specific starch source can have an influence on larval performance, rather than the absolute amount of starch [27]. Indeed, mealworms can utilize a wide range of carbohydrates, aside from cellulose or lactose. Lethal effects in mealworms can also occur because of essential oils (e.g., in garlic) which are harmful to insects [22], acting as an insecticide. Many insect species are affected by methylxanthines (including caffeine). These substances can inhibit food consumption and caffeine is moreover known to be a naturally occurring pesticide [34]. It is thus not surprising that one of the least consumed ingredients in a self-selection experiment was coffee chaff [26]. In our study, larvae fed with CG were found on top of the substrate and even tried to escape from the box. The first dead individuals were recorded after three days, whereas all individuals of the control group (without any substrate) were vital after one week of observation. The substrates leading to no biomass growth in this study generally contained high amounts of fiber (CG, WC, CC, or Sida with a crude fiber content of 26–61% in DM, Figure 11) at the expense of other nutrients or have a combination of high fiber and high protein contents like soybeans [26]. High-fat diets such as PSO promote the potential agglomeration of the substrate resulting in lower aeration and thus negatively interfering with respiration and movement of mealworms [35]. After three days of observation larvae reared on PSO seemed lethargic (upside down in the box), although not a single individual was found dead after one week of daily observation. 

The most limiting growth factor concerning macronutrients—also in this study—is the carbohydrate content of the substrate [1,2,22,26]. Another limitation for development and survival—enhancing the growth rate and highly influencing the life cycle of *T. molitor*—is the protein content [1,2,3,5,14]. An optimal ratio of 50:50 was proposed [36] and in a study, larvae were fed with 10:90 parts of yeast and whole ground wheat, leading to a much higher weight gain [5]. Moreover, supplementation with protein increased survival rates from 19–52% to 67–79% [3], from 84–88% to 88–92% [2], and from 13–14% to 22–23%, respectively [3]. In general, diets with high protein content (30–39% dry mass) reduce the time for pupation from 191–227 days to 116–144 days (28 °C, 70% RH, [3]) and the time to 50% pupation from 95–168 days to 79–95 days (28 °C, 65% RH, [2]). Larvae fed diets with low contents of protein did not reach the pupal stage after one year of rearing [1]. Adding lipids to the diet is beneficial at low concentrations (cholesterol is a necessary ingredient); higher concentrations (>3% cholesterol), however, become an inhibitory factor and can be harmful [13,14]. The highest fat content concerning “successful” substrates in this study amounted to 11% (CGo).

Regarding this study, the most successful substrate was MRP (malt residual pellets) composed of 29% starch, 19% protein, 3% fat, and 8% sugar (Figure 11). Similar ratios of macronutrients (carbohydrates, proteins, lipids) were observed, producing the best growth performance for *T. molitor* larvae [22,26]. The optimal ratio was accordingly 9–10% for lipids, 20–23% for proteins, and 67–72% for carbohydrates [22], compared to 6% lipids, 23% proteins, and 71% carbohydrates [26]. To increase the complexity, it was shown that younger larvae need more carbohydrates for growth than older ones and protein and fat requirements grow with increasing age [14,22]. This can be explained by the fact that the larvae are preparing for the energy intensive metamorphosis and primarily lipids and proteins are used as an energy supply. Moreover, after mealworms reach half- or full-grown size, no addition of vitamins is required to complete larval development and pupation [5].

*Tenebrio molitor* is further sensitive to many environmental factors such as the temperature or the larval density, influencing different growth parameters. A larval weight of 0.40 g increased up to 0.77 g at low rearing densities [37], or larvae at 25 °C weighed 0.12 g compared to 0.36 g at 30 °C [19]. Individual larval weight is thus variable based on the study; the final weight of larvae fed with different diets, for example cookies, bread, and a mixture of bread and cookies, and brewer’s spent grain and cookies (50:50), was 0.09 g, 0.10 g, 0.11 g, and 0.17 g, respectively [1]. The average larval weight for specimens reared on a high protein concentration diet amounted to 0.14 g [2]. The heaviest larva fed with 100% wheat bran weighed 0.07 g [38], compared to an average larval weight of 0.20 g (fed with the same substrate), and a maximum larval weight of 0.21 g (wheat bran and distillers dried grain) [19] and 0.22 g fed with 70% WB and 30% brewer’s spent grain [33]. The maximum individual weight of a larva fed with wheat bran in our feeding trail was 0.18 g. The pupal weight ranged from 0.13 to 0.15 g [12] and a maximum pupal weight was recorded for pupae fed with wheat bran amounting to 0.11 g [38]. Concerning our study, the maximum pupal weight amounted to 0.24 g (WBc, Table 1).

Regarding economic facts, the costs of substrates for *T. molitor* are a key issue [23]. The major expenses in rearing are related to acquiring raw materials for feeding the insects [23]. The use of by-products can greatly reduce the costs of rearing as those substrates are cheaper than conventional feed and are easy to obtain from the industry in high amounts. Costs and labor can also be reduced by minimizing the manipulation of the substrate by means of drying, cutting, grinding, et cetera. Many substrates, also in this study (cf. Appendix A), had to be dried, cut, or ground before use. Also, adding supplements should be kept at a minimum in order to maintain low costs [1].

Another hurdle in the European Union concerning the selection of the substrate is the fact that insects for human consumption have to meet food safety regulations; thus, specimens can only be fed with specific, approved diets of (mainly) vegetable origin [23].

Using by-products and leftovers for *T. molitor* rearing can lead to a decrease in waste materials and pollution and represents an interesting opportunity in terms of a circular economy [1,28]. Although, when choosing a substrate there is a big trade-off among many variables which can be altered by feed and must be taken into account.

## 5. Conclusions

As the human population and consequently global demands are growing rapidly, a protein source with low environmental impact is needed. In the course of this study, we were able to identify eight single substrates (wheat bran, malt residual pellets, corn germ meal, sweet chestnuts, sweet potatoes, wheat germs, soybeans, and bread remains) which are suitable for the mass rearing of *T. molitor.* This study clearly demonstrates that mealworms can be reared on single, sustainable by-products, leading to a cost-effective protein source with low environmental impact. However, further research should be undertaken by analyzing the nutrient composition of differently fed *T. molitor* larvae as it has been discovered that diet can modify the nutritional profile of the larvae.

## Figures and Tables

**Figure 1 foods-12-04092-f001:**
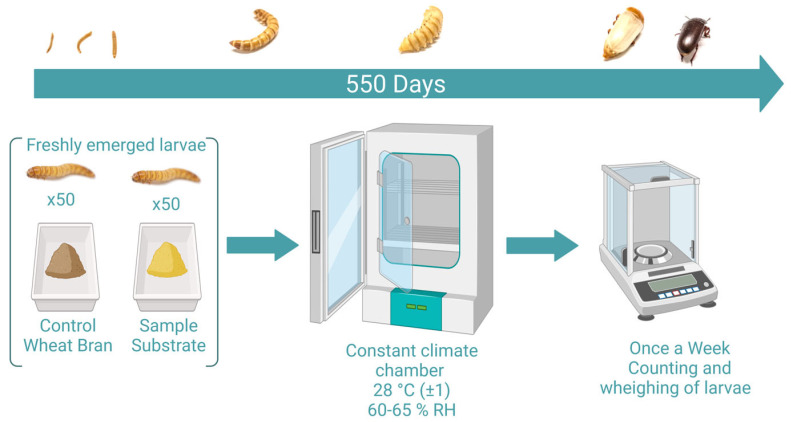
Setup of the feeding experiments. This graph was created with BioRender.com.

**Figure 2 foods-12-04092-f002:**
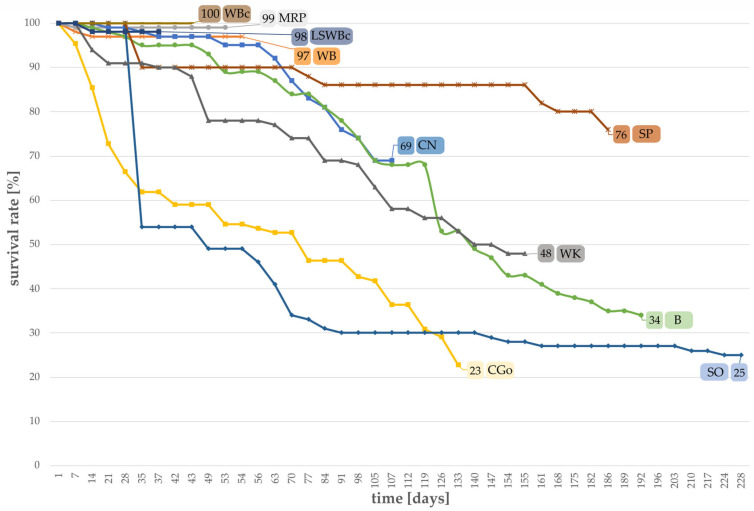
Larval survival compared among different successful substrates, determined daily, and given in percent until the first pupa was observed (WB, MRP, CGo, CN, B, SO, WK: n = 100; LSWBc, WBc, SP: n = 50).

**Figure 3 foods-12-04092-f003:**
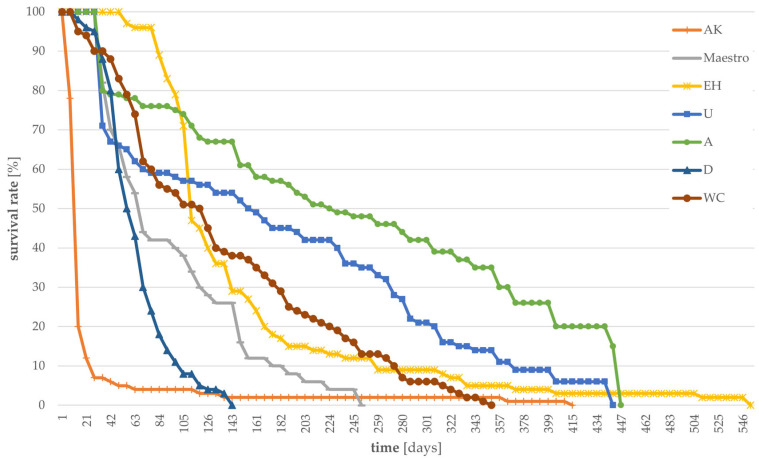
Larval survival in percent over time, determined daily, for different “improvable” substrates. (n = 100 for AK, EH, U, A, D, WC, and n = 50 for Maestro).

**Figure 4 foods-12-04092-f004:**
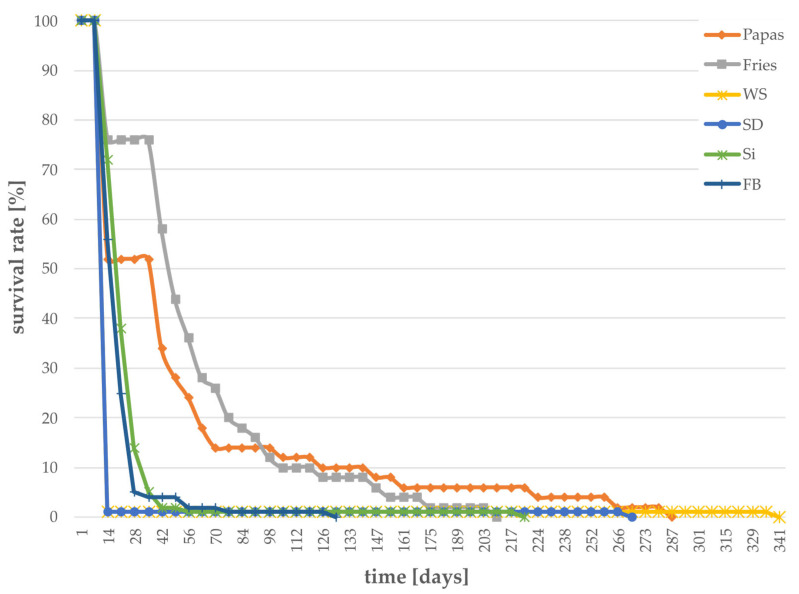
Larval survival in percent for different “not recommended” substrates with data for larvae surviving > 200 days. (n = 100 for WS, SD, Si, FB, and n = 50 for Papas, Fries). Survival was determined daily until all larvae were dead.

**Figure 5 foods-12-04092-f005:**
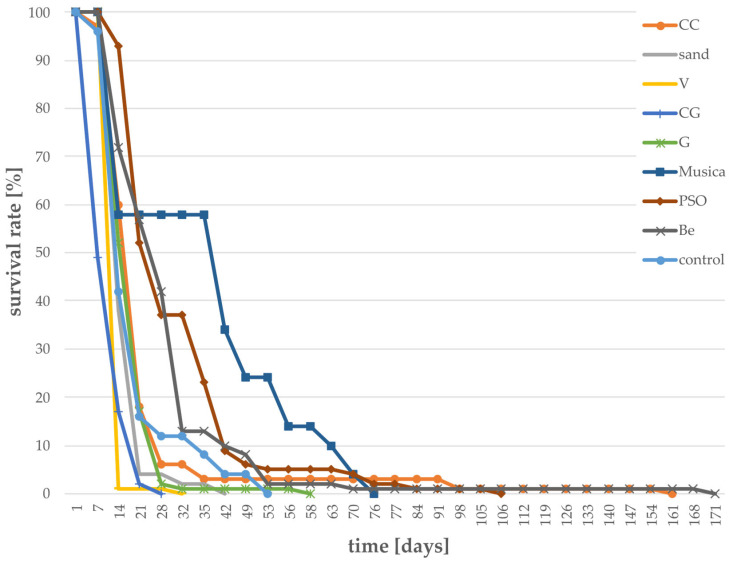
Larval survival in percent for different “not recommended” substrates (survival of <200 days), n = 100 for CC, CG, G, PSO, Be, and n = 50 for sand, V, Musica. Survival was determined daily until all larvae were dead.

**Figure 6 foods-12-04092-f006:**
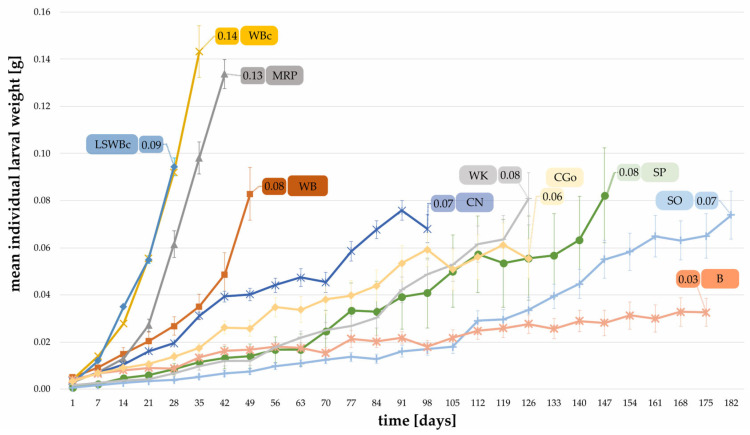
Mean individual weight of larvae (determined weekly) reared on different substrates shown until the first pupa emerged. (WB, B, SO, WK, CGo, CN, MRP: n = 20; SP, LSWBc, WBc, n = 10).

**Figure 7 foods-12-04092-f007:**
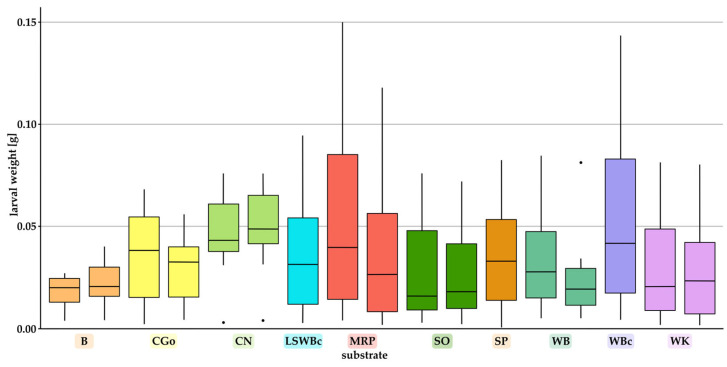
Boxplot of Kruskal–Wallis test, comparing different mean values (based on the individual larval weight) among investigated substrates until the first pupa emerged (WB, B, SO, WK, CGo, CN, MRP: two independent boxes; SP, LSWBc, WBc: one box). The dots represent outliers.

**Figure 8 foods-12-04092-f008:**
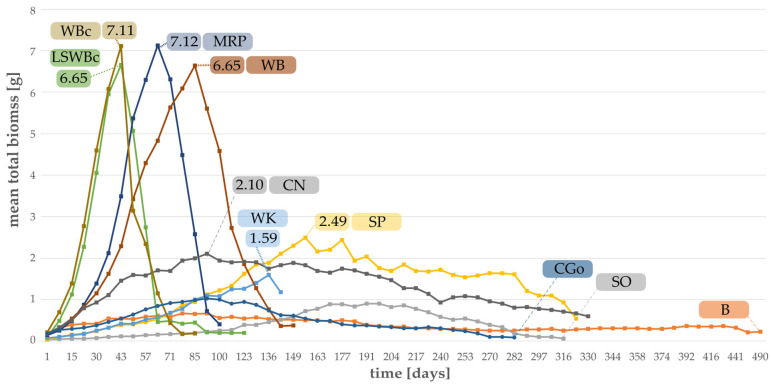
Mean total biomass of larvae grown on “suitable substrates” given in gram, determined weekly. Numbers indicate peaks of mean total biomass given in gram for seven most successful substrates (MRP, WB, CN, WK, SP, B, CGo, SO) and two control diets (WBc, LSWBc). For CGO, SO, and B, no clear peak is visible.

**Figure 9 foods-12-04092-f009:**
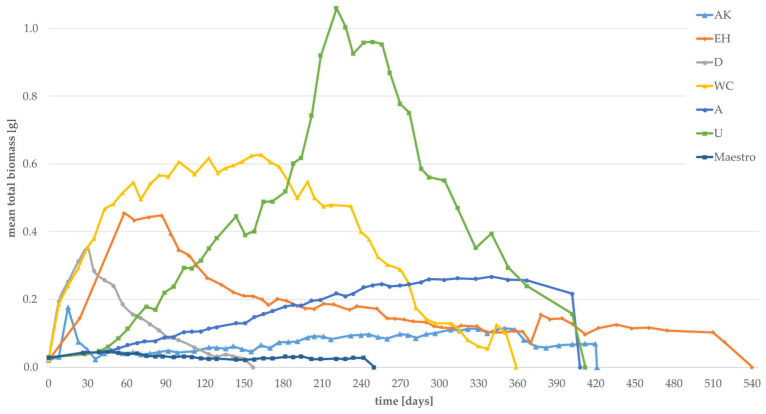
Mean total biomass of “improvable substrates” in gram, determined weekly.

**Figure 10 foods-12-04092-f010:**
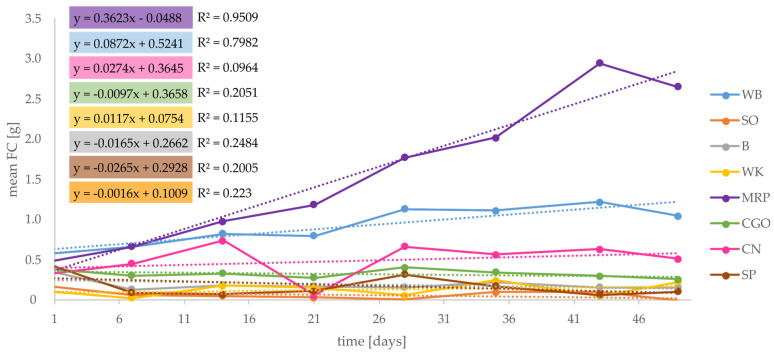
Mean FC (feed consumed) for “successful” single substrates (WB, SO, B, WK, MRP, CGo, CN, and SP) in gram (n = 50).

**Figure 11 foods-12-04092-f011:**
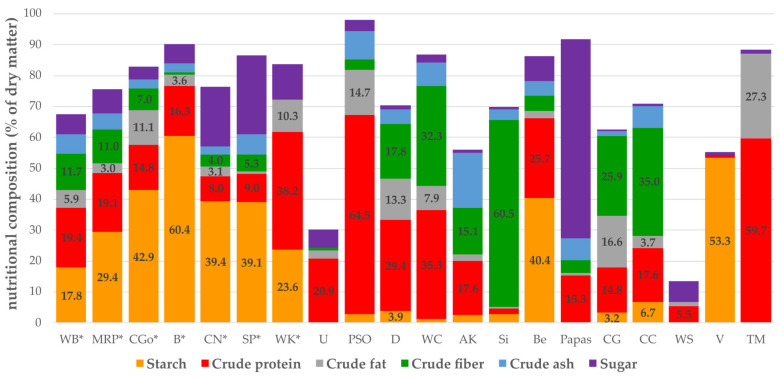
Nutrient composition of substrates in percent of dry matter (g/kg DM) and *Tenebrio molitor* larvae (TM = fed with wheat bran and carrots). * Substrates which were “successful” in feeding trials. Starch, protein, and fat content <3% are not indicated as number in this graph.

**Table 1 foods-12-04092-t001:** Days until occurrence of the first pupa (development time), pupae mean, maximum, minimum weight, and standard deviation, sex ratio (males and females), pupation rate (percentage of emerged pupae), and percentage of emerged beetles per substrate.

Diet	1st Pupa ^1^ [Days]	Pupation Rate ^2^ [%]	Mean Pupal Weight ±SD [g]	Min/Max Weight [g]	Sex Ratio m/f [%]	Emerged Beetles ^3^ [%]
CGo	132	3	0.13 ± 0.02	0.11/0.15	33/67	3
CN	106	14	0.12 ± 0.04	0.04/0.19	43/57	12
MRP	52	93	0.16 ± 0.02	0.11/0.21	49/51	95
SP	186	20	0.11 ± 0.01	0.10/0.13	50/50	20
WB	53	79	0.15 ± 0.02	0.10/0.21	45/55	73
LSWBc	73	98	0.15 ± 0.02	0.10/0.23	53/47	98
WBc	42	96	0.17 ± 0.03	0.12/0.24	47/53	98
WK	154	34	0.11 ± 0.02	0.06/0.14	41/59	20
B	191	3	0.11 ± 0.00	0.11/0.12	67/33	0
SO	227	2	0.11 ± 0.01	0.10/0.12	50/50	0

^1^ counted from the start of the feeding trail, ^2^ % of total number of investigated larvae, ^3^ based on the total number of pupae.

**Table 2 foods-12-04092-t002:** Mean, minimum, maximum weight, and standard deviation of individual beetles, duration of pupal stage (mean, min, max), and morphological anomalies in percent of beetles.

Diet	Mean Weight ±SD [g]	Min/Max Weight [g]	Duration of Pupal Stage [days] (Min/Max)	Morph. Abnormalities ^1^ [%]
CGo	0.11 ± 0.02	0.08/0.14	7 (7/8)	33
CN	0.10 ± 0.04	0.07/0.16	7 (4/10)	83
MRP	0.14 ± 0.02	0.09/0.19	6 (5/8)	22
SP	0.10 ± 0.01	0.07/0.11	6 (4/7)	40
WB	0.13 ± 0.02	0.07/0.17	7 (4/10)	32
WK	0.10 ± 0.01	0.09/0.13	6 (5/7)	0
LSWBc	0.13 ± 0.02	0.08/0.21	6 (4/7)	39
WBc	0.14 ± 0.02	0.10/0.20	7 (5/9)	16

^1^ observed abnormalities in wings and elytra (deformed, folded), legs, antennae, segments (uncovered abdomen), and other structures of emerged beetles.

**Table 3 foods-12-04092-t003:** Overview of investigated substrates and suitability for rearing and feeding of *T. molitor*.

Substrate	Abbreviation	Suitability	
wheat bran	WB	✓	main rearing substrate
malt residual pellets	MRP	✓	
corn germ meal	CGo	✓	
sweet chestnuts with peel	CN	✓	
sweet potatoes	SP	✓	
wheat germs (extracts)	WK	✓	
soybeans	SO	✓~	main feeding substrate ^1^
bread remains	B	✓~	
brewer´s spent grain	D	~	supplement
hempseed cake	WC	~	
*Urtica*	U	~	
acron flour	EH	~	
pearl oyster mushroom	A	~	
potatoes	Maestro	~	
foam peanuts	V	~	
pearl oyster mushroom mycelia with coffee grounds and coffee chaff	AK	~	
coffee grounds	CG	✕	not suitable
coffee chaff	CC	✕	
pumpkin kernel cake	PSO	✕	
*Fallopia* x *bohemica*	FB	✕	
*Sida*	Si	✕	
garlic peel	G	✕	
runner beans	Be	✕	
Mur sand	Sand	✕	
sawdust	SD	✕	
wheat straw	WS	✕	
potatoes	Musica	✕	
potatoes	Fries	✕	
jerusalem artichoke	Papas	✕	

^1^ not suitable as single substrate for breeding purposes.

## Data Availability

The data used to support the findings of this study can be made available by the corresponding author upon request.

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
