# Peer review of "Future Proteins: Sustainable Diets for Tenebrio molitor Rearing Composed of Food By-Products"

_foods, 2023, doi:10.3390/foods12224092_

Round 1
Reviewer 1 Report
Comments and Suggestions for Authors
The work is framed in the use of different raw materials for the production of protein from the tenebrio molitor insect.
The article is well structured, however there are a number of opportunities for improvement:
In l145 it talks about the use of an amount of substrate between 2-18 grams depending on its texture, shouldn't it be depending on its density?
In l157-158 the comment regarding the absence of water and mortality is unnecessary.
In the feeding experiments were the insects only weighed or were they also measured?
In l179 you speak of "potatoes", do you mean potatoes?
It would be very useful and clarifying to add a table with the different diets given to the insects. In the same vein, it would be interesting to provide information on their chemical composition in general terms (macronutrients).
In l209 it is mentioned about analysis of the substrates but it is not clear what exactly was done.
Figures 2, 5 and 6 have no legend.
The time scales for graphs measuring the same variable need to be revised and standardised.
The analysis of the different curves is very simple, it would be interesting to evaluate the possibility of modelling with some mathematical equation to have a deeper analysis and to be able to compare better.
better compare.
l365 review etc.
In some parts of the discussion it talks about composition but there is no item that contains these results...? do they come from chemical analysis or are they theoretical information from tables?
It is striking that the work does not contain a section of conclusions where the contribution of the work to the discipline and its possible applications are presented.
Reviewer 2 Report
Comments and Suggestions for Authors
The article is interesting and concerns current issues. However, I believe it requires a several changes.
The abstract should be re-written to focus more on the results and conclusions obtained. Currently, half of the abstract is a theoretical introduction.
Line 33: Fatty acid profiles determined in many studies do not indicate a particularly favorable fatty acid composition in TM. It is a fat quite comparable to other animal fats. In my opinion, it is difficult to call it well-balanced in the context of human nutrition.
The materials and methods section lacks information about the chemical analyzes performed in the experiment.
There is also no information about statistical analysis.
Fig 2 Why did some experiments last 228 days (SO) and some about 50 days (WBc; WB; LSWBc)?
The charts are prepared carelessly. Why no attempt was made to fit some model to the data (except for Fig. 8). The lack of statistical analysis in charts and tables makes it impossible to draw conclusions.
Chemical analyzes must be described precisely, indicating the method, equipment and method of sample preparation. A number of doubts arise regarding the results regarding protein, which, due to the title of the article, is the most important component. What method was the protein determined (probably Kjeldahl method). How then were the results confounded by the presence of chitin?
Why were insects weighed to two decimal places? Wouldn't it be better to weigh them to four decimal places? For such small objects, the accuracy is too low.
Reviewer 3 Report
Comments and Suggestions for Authors
1. I would recommend slightly changing the abstract - the reader should know your main findings from the abstract itself, e.g. you can include which by-products were detected as the best ones for rearing the larvae.
2. Line 138 - "for about three weeks" - Personally, I would recommend providing the exact time since based on your methodology the experiment should be repeatable.
3. Line 150 - ad libitum should be written in italic.
4. Lines 183-214 - For better clarity of subchapter 2.2 it would be better to use different text hierarchy - personally, I would suggest moving the explanations under the corresponding values.
5. In subchapter 2.2. "data analysis" you don't mention any statistical tests - have you tried to express obtained results also from a statistical point of view? I would strongly recommend doing that.
6. I think it would be good to include at least a citation of a methodology used for chemical analyses of used substrates (I know the analyses were done at different workplaces, but I also think it is quite important for the reader to know which methods were used).
7. Figure 4 - I would suggest dividing these figures because of different scales on their x axes. As a reader, this layout makes me want to visually compare them, but that cannot be done with different scales. There's also the option of unifying the scales, but in case of your results it would probably worsen the clarity of the figure.
8. Line 275-276, line 290, ... - Please keep in mind that value and its unit should be on the same line.
9. I would suggest adding a "conclusion" chapter, where you can briefly summarize your main findings. Both chapters "results" and "discussion" are a bit long and I believe that adding "conclusion" could help your readers to easily understand your study, to remember your key results and perhaps to use them and cite them in the future.
I find your study interesting and I think it would be also beneficial to analyze nutrient composition of differently fed T. molitor larvae as it has been discovered that altered diet can modify final nutritional profile of the larvae and afterwards they can be utilized in various kinds of final food products. Do you plan to do any experiments like this?
Comments on the Quality of English LanguageSome of the sentences are a bit hard to follow or they can be improved both in terms of stylistics and grammar etc., e.g.:
line 21 - the last sentence of your abstract is hard to follow
lines 26-29 - these two sentences are really similar (almost the same) and can be merged into one
lines 26-39 - overuse of "grow", sometimes it doesn't really fit the meaning (line 36)
line 69 - "carry out" - please refrain from using phrasal verbs as the common goal is make academic text understandable for everyone even with basic knowledge of English
line 110 - "imported"
line 124-127 - the aim of the study should be easy to follow, I would suggest dividing the sentence
line 135 - "maintain" - try to look for more suitable synonyms
line 137 - "in" - perhaps it would be better to use "on"
line 150 - I agree that "tested" can be used with "substrates", but it doesn't make sense with "experiments"
line 151 - try to think about other ways to express "diet", in this case I don't find it very suitable - perhaps "substrates" or "feed" would be better, this problem also occurs in different parts of the manuscript (e.g. line 153)
line 179 - "were ground by means of" please revise the whole expression - consider simplifying it (line 183 - the same problem)
line 219 - "yielding in" incorrect use of preposition
I have summarized a few examples for you to know what can be improved, but I would suggest going through the whole manuscript and revising it. There are several ways to improve the text flow and stylistics and it could help the reader to understand everything easily. Sometimes it would be enough to change the word order in a sentence, e.g. line 37 - "economically", but in other cases a revision of whole sentence is needed. Of course a big part of the manuscript is easy to follow and read, but there are parts which can be improved.
Regarding grammar - I have not detected many major problems, but please check especially the correct use of verbs and prepositions (some of the problems are mentioned above).
Round 2
Reviewer 1 Report
Comments and Suggestions for Authors
The authors responded satisfactorily to most of the comments and suggestions. The quality of the work has improved considerably and it is recommended that the article be approved in its present form.
Author Response
Thank you very much for your comments and suggestions to improve our manuscript.
Reviewer 2 Report
Comments and Suggestions for Authors
Unfortunately, the authors did not implement the suggested changes. In particular, in the methodological part there is still no description of analytical methods, which is an unacceptable error in experimental work.
Author Response
A description of the analytical methods is provided, and statistical analysis is now included as part of the article.
Reviewer 3 Report
Comments and Suggestions for Authors
Thank you for considering all suggestions and making all the changes within your manuscript. Finally, just a minor final suggestion - you decided to include statistical data in the supplementary materials. Personally, I would rather include it in the article, but in the end, this is up to you and the editor, the important thing is that it is included.
Author Response
Thank you very much for your constructive and helpful comments and suggestions. Statistical analysis is now included as part of the article (Figure 7).